# Review of Respirable Coal Mine Dust Characterization for Mass Concentration, Size Distribution and Chemical Composition

**Behrooz Abbasi [1,\*]**, **Xiaoliang Wang [2,\*]**, **Judith C. Chow [2]**, **John G. Watson [2]**, **Bijan Peik [3]**, **Vahid Nasiri [4]**, **Kyle B Riemenschnitter [5]** and **Mohammadreza Elahifard [1]**

1.  Department of Mining and Metallurgical Engineering, University of Nevada, Reno, NV 89557, USA; Melahifard@unr.edu
2.  Division of Atmospheric Sciences, Desert Research Institute, Reno, NV 89512, USA; Judy.Chow@dri.edu (J.C.C.); John.Watson@dri.edu (J.G.W.)
3.  Golder Associates Inc., Toronto, ON M5H 3R3, Canada; Bpeik@golder.com
4.  College of Business, University of Nevada, Reno, Reno, NV 89557, USA; Vnasiri@unr.edu
5.  Department of Mechanical Engineering, University of Nevada, Reno, Reno, NV 89557, USA; Kriemenschnitter@unr.edu
\*  Correspondence: Abbasi@unr.edu (B.A.); Xiaoliang.Wang@dri.edu (X.W.); Tel.: +775-784-6907 (B.A.); +775-674-7177 (X.W.)

**Abstract:** Respirable coal mine dust (RCMD) exposure is associated with black lung and silicosis diseases in underground miners. Although only RCMD mass and silica concentrations are regulated, it is possible that particle size, surface area, and other chemical constituents also contribute to its adverse health effects. This review summarizes measurement technologies for RCMD mass concentrations, morphology, size distributions, and chemical compositions, with examples from published efforts where these methods have been applied. Some state-of-the-art technologies presented in this paper have not been certified as intrinsically safe, and caution should be exerted for their use in explosive environments. RCMD mass concentrations are most often obtained by filter sampling followed by gravimetric analysis, but recent requirements for real-time monitoring by continuous personal dust monitors (CPDM) enable quicker exposure risk assessments. Emerging low-cost photometers provide an opportunity for a wider deployment of real-time exposure assessment. Particle size distributions can be determined by microscopy, cascade impactors, aerodynamic spectrometers, optical particle counters, and electrical mobility analyzers, each with unique advantages and limitations. Different filter media are required to collect integrated samples over working shifts for comprehensive chemical analysis. Teflon membrane filters are used for mass by gravimetry, elements by energy dispersive X-ray fluorescence, rare-earth elements by inductively coupled plasma-mass spectrometry and mineralogy by X-ray diffraction. Quartz fiber filters are analyzed for organic, elemental, and brown carbon by thermal/optical methods and non-polar organics by thermal desorption-gas chromatography-mass spectrometry. Polycarbonate-membrane filters are analyzed for morphology and elements by scanning electron microscopy (SEM) with energy dispersive X-ray, and quartz content by Fourier-transform infrared spectroscopy and Raman spectroscopy.

**Keywords:** respirable coal mine dust; black lung; silicosis; size distribution; chemical composition

## 1. Introduction

Inhalation of respirable coal mine dust (RCMD) particles (with aerodynamic diameters $\lesssim$4 micrometers [μm]), and especially those containing quartz (crystalline silica), has been associated with coal workers' pneumoconiosis (CWP, sometimes referred to as "black lung") and silicosis diseases [1]. The extent, intensity, and constituents of RCMD exposure have been directly related to risks of human lung cellular damage and inflammation [2].

Implementation of the Mine Health and Safety Act of 1969 resulted in the reduction of RCMD mass and crystalline silica concentrations in U.S. mines [3]. The National Institute of Occupational Safety and Health (NIOSH) [3] reported corresponding decreases in CWP occurrences for mid-central and south-central Appalachia underground coal miners between 1970 and 2000 (Figure 1). The 1969 regulation, along with improved mine ventilation, has resulted in reducing workplace disease [4,5]. Since 2000, however, the prevalence and severity of RCMD-related lung diseases have increased [6,7], especially in mid-central Appalachia. New CWP and/or silicosis diagnoses are appearing in younger miners who should have benefitted from mine safety regulations [8–11]. The 2014 Mine Safety and Health Administration's (MHSA) [12] respirable coal dust rule reduced permissible RCMD exposure from 2.0 to 1.5 mg/m$^3$ over a full work shift. As a result, respirable dust sampling has gained importance for quantifying worker exposures and identifying RCMD sources. To improve measurement quality, MSHA [12] further requires the use of an approved continuous personal dust monitor (CPDM) from 2016 to complement the coal mine dust personal sampler unit (CMDPSU). As a comparison, the United States Environmental Protection Agency's (U.S. EPA) [13] national ambient air quality standards (NAAQS) for maximum 24 h PM$_{2.5}$ (particles <2.5 μm aerodynamic diameter) exposure is 0.035 mg/m$^3$.

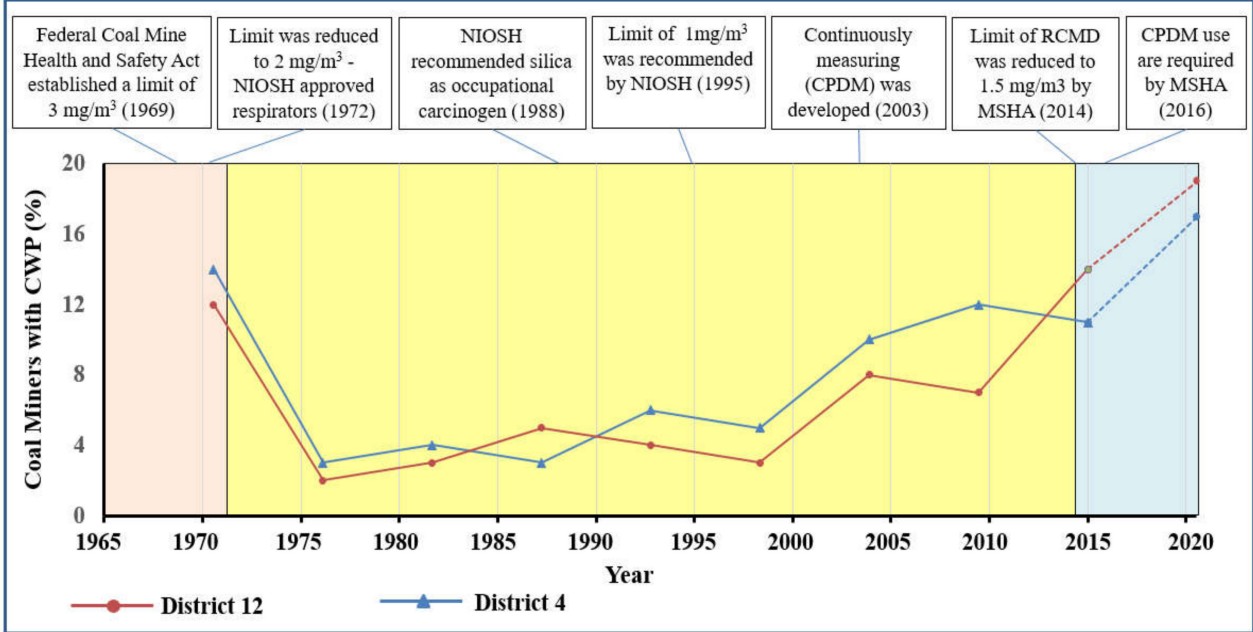

**Figure 1.** Coal workers' pneumoconiosis (CWP) prevalence in mid-central (District 4, Southern West Virginia) and south-central Appalachia (District 12) underground coal miners between 1970 and 2014. Data acquired from the Coal Workers' Health Surveillance Program (CWHSP) data query system [3,14], and includes all reported categories of CWP. CPDM: continuous personal dust monitor.

RCMD properties other than mass and crystalline silica, such as size, morphology, and chemical composition, also affect human health. Inhaled dust is deposited in different regions of the respiratory tract depending on particle sizes and shapes. The American Conference of Governmental Industrial Hygienists (ACGIH) established the following size fractions: (1) inhalable–particles capable of entering the nose and mouth; (2) thoracic–particles penetrating beyond the larynx; and (3) respirable–particles penetrating to the gas exchange region (alveolar) of the lung [15]. Size-selective sampling of these size fractions is defined by particle penetration efficiency curves, with 50% efficiencies at ~100, 10, and 4 μm, respectively. In underground coal mines, dust particles ≥1 μm aerodynamic diameter from mechanical processes dominate the particle mass; however, ultrafine particles <0.1 μm can dominate particle number concentrations in the presence of diesel engine ex-

haust [16]. Ultrafine particles present a health threat because of their potential to penetrate deep into the lung and pass across the air–blood barrier. Ultrafine particles present large surface areas that promote reactions with body fluids [17–19].

Mine safety regulations require RCMD mass to be collected by a size-selective cyclone inlet with a cut-off diameter (50% penetration efficiency) of ~4 µm [20], which is an approximation of the inhalation properties of the human lung [21]. However, depending on particle size, RCMD mass can differ from the amount of dust that would deposit (i.e., dose) in the lung [22]. Size distribution measurements spanning the range from <0.1 µm to >1 µm particles are needed to assess potential health effects. Continuous size distributions enable the evaluation of metrics such as mass, surface area, and number concentration relationships to adverse health effects. Size distributions are also relevant to effective emission reduction measures, flammability, and explosive potential [23].

Most coal mine dust size distributions were collected over a decade ago [24,25], and may no longer be representative due to changes in underground coal mining conditions and practices. As coal seams become thinner, more rock strata (immediate roof and floor) are mined. Advances in longwall shearers have increased the volume of material handling, which can increase coal mine dust generation. It is important to understand how these changes affect particle size, shape, concentration, and composition. The National Academies of Sciences, Engineering, and Medicine (NASEM) [22] recommended a comprehensive characterization of RCMD chemical compositions and size distributions to identify additional causes of lung disease.

Depending on mine geology and mining practices [26], RCMD consists of different chemical components with varying toxicities. Crystalline silica and diesel particulate matter (DPM), classified as Class I human carcinogens by the International Agency for Research on Cancer (IARC), are found in underground coal mines and cause excess lung cancer mortality [22]. RCMD contains transition metals such as iron (Fe), vanadium (V), nickel (Ni), chromium (Cr), and copper (Cu), which have the potential to generate reactive oxygen species (ROS) in biological tissues. Other non-redox active metals, such as zinc (Zn), aluminum (Al), and lead (Pb), can exacerbate RCMD toxicity [17]. Some organic compounds in mine dust, such as polycyclic aromatic hydrocarbons (PAH) and their derivatives are designated as hazardous air pollutants that may cause cancer [27]. A comprehensive characterization of RCMD chemical composition can be used to identify its sources and evaluate health effects that will lead to more effective mitigation strategies.

This review identifies and summarizes literature regarding methods to quantify RCMD mass concentrations, size distributions, chemical compositions, and minerology relevant to coal mines. This review focuses on airborne RCMD and samples collected on filter media. Characterization of RCMD from lung tissue biopsies is not reviewed here [28,29].

## 2. RCMD Mass Measurement Methods

Table 1 compares three commonly applied technologies for RCMD mass concentration measurement. Mine safety regulations require personal exposure monitoring in miners' breathing zones. In the U.S., RCMD mass concentrations are conventionally determined by sampling with a CMDPSU onto a filter followed by gravimetric analysis in a laboratory [30]. As shown in Figure 2a, the CMDPSU is equipped with a belt-mounted constant-flow pump that draws air at 2 L per minute (L/min) through a 10-mm nylon Dorr–Oliver cyclone (or equivalent) and a pre-weighed filter. Under this flow rate, large particles with an aerodynamic diameter ($d_{ae}$) > 10 µm are removed and collected in the cyclone hopper, which is cleaned between each use. Penetration efficiencies are ~50% for particles with $d_{ae} \approx 4$ and 100% for $d_{ae}$ < 2 µm [31]. As the cyclone sampling effectiveness curve [32] varies with flow rate, empirical conversion factors are applied to compensate for these changes. Due to the differences between the cyclone penetration- and respirable dust deposition-efficiency curves, conversion factors are also used to convert CMDPSU concentrations to other respirable dust conventions, such as the British Medical Research Council (BMRC) and the

International Organization for Standardization (ISO) definitions [33]. Downstream of the cyclone, particles are collected onto a 37-mm-diameter polyvinyl chloride (PVC) filter with a pore size ≤5 μm. The filters are sent to a laboratory for gravimetric and sometimes crystalline silica measurements.

**Table 1.** Comparison of RCMD mass concentration measurements.

| Method | Description | Limitations and Challenges |
|---|---|---|
| Gravimetric sampler | Constant-flow sampling through a particle size-selective cyclone (e.g., Dorr–Oliver) onto a filter cartridge by a personal sampling pump<br>The filter is submitted to gravimetric analysis and optionally for chemical analysis in the laboratory<br>Reference method<br>Relatively low cost | Ensuring that the cyclone assembly stays upright<br>Labor intensive<br>Low time resolution<br>Data are not immediately available |
| Continuous personal dust monitor (CPDM) | A TEOM (tapered-element oscillating microbalance) obtains near real-time, gravimetric-equivalent measurement of RCMD mass concentrations<br>Filter can be used for limited laboratory analysis<br>Near real-time measurement (30-min average)<br>Regulatory requirement<br>Relatively independent of aerosol optical, physical, and chemical properties | High cost<br>Size and weight are burdensome<br>Regulatory requirement to report data to MSHA<br>Potential evaporation losses |
| Photometer | Inferred mass concentration based on aerosol light scattering intensity<br>Low cost<br>Lightweight<br>Fast response (~1 s) | Scattering-mass relationship varies with particle refractive index, shape, size distribution, density, and relative humidity<br>Field calibration is needed |

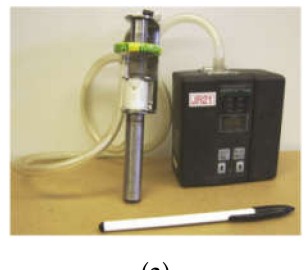 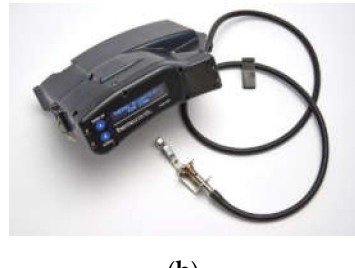 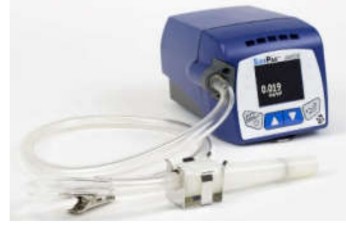

(**a**)     (**b**)     (**c**)

**Figure 2.** Examples of RCMD mass measurement devices: (**a**) a coal mine dust personal sampler unit (CMDPSU) with a sampling pump, cyclone, and filter cassette; (**b**) ThermoScientific Personal Dust Monitor (PDM) 3700; and (**c**) TSI SidePak photometer personal aerosol monitor (SidePak has not been approved by the MSHA [Mine Safety and Health Administration] for use in underground coal mines).

Filter sampling and gravimetric analysis [34] has been used as a reference method to demonstrate compliance with the RCMD exposure limit. However, it has several shortcomings. First, RCMD concentrations may take days or weeks to obtain, failing to provide critical information about the causes or prevention of overexposure. Second, the filter sample is collected over the entire shift and does not record temporal variations of RCMD exposures. Third, particle accumulation on cyclone walls, electrostatic charges, and cyclone orientation may affect the cyclone performance and introduce measurement bias [35–37]. The cyclone assembly must remain upright with the hopper facing downward, a stance difficult to maintain for the range of job activities. If the cyclone orientation is altered during measurement, oversized dust particles can deposit onto the filters, creating impurities that lead to inconclusive or inaccurate results.

Since February 2016, the use of a real-time CPDM for occupational exposure in high concentration areas and for miners with symptoms related to the development of pneumoconiosis has been required [30]. The CPDM continuously measures RCMD mass concentrations and reports within-shift (30 min running average) and end-of-shift concentrations promptly upon the completion of the work shift [30]. If RCMD concentrations exceed the permissible exposure limit, the mine operator is required to take immediate corrective actions. Presently, the only approved commercial CPDM is the personal dust monitor (PDM 3700, Thermo Fisher Scientific, Franklin, MA, USA; Figure 2b) [38]. The PDM 3700 and its predecessor, PDM 3600, use a tapered-element oscillating microbalance (TEOM) to continuously measure the mass of collected particles [39]. Particle-laden air is drawn through an inlet positioned in the miner's breathing zone at a flow rate of 2 L/min. The respirable dust is size-classified by a Higgens–Dewell type cyclone [31,40] and transported through a heated transfer line to the mass transducer worn at the miner's waist. Particles are collected on a Teflon-coated glass-fiber filter mounted on top of an oscillating hollow tapered element, for which the frequency decreases as particles deposit on the filter. The relationship between mass and tapered-element frequency changes is determined from calibration with known masses [41]. The TEOM technology has been widely used in ambient particulate matter (PM) monitoring and is designated as a federal equivalent method by the U.S. EPA [42]. The PDM is a miniaturized version created for mining applications.

Laboratory and field measurements have evaluated PDM performance. Volkwein et al. [41,43] evaluated prototype PDMs in the laboratory using resuspended coal dust, finding that PDM mass concentrations were within ±25% of reference gravimetric measurements. Further field tests in ten coal mines found that the PDM had ~90% valid data availability for over 8000 h of underground use. Page et al. [44] conducted a linear regression of 129 pairs of PDM and CMDPSU measurements from 180 mechanized underground coal mining units and found the regression slope to be 1.05 (with zero intercept). Laboratory studies demonstrate that the PDM compares favorably with gravimetric mass concentrations for different aerosols, such as wood dust, aluminum oxide powder, flour dust, grain dust, diesel exhaust, welding fumes, Arizona road dust, and sodium chloride [45–47]. However, several studies indicate that transport losses and particle blow-off from the PDM filter may underestimate concentrations [45]. Loss of volatile material (as in diesel exhaust) due to the heating of the air inlet and tapered element to ensure stability may also result in negative biases for mass concentration.

The main PDM advantage is that it is comparable to gravimetric measurements and its response is independent of aerosol refractive index, size distribution, and density. The near real-time measurement provides miners with timely information to identify factors contributing to overexposures, allowing corrective actions to be taken immediately. In addition, the CPDM filters can be submitted to a laboratory for some chemical analyses [48,49]. However, its cost, size, and weight are drawbacks for routine use [22]. The high cost (~US$17,000) limits the number of instruments used for purposes other than regulatory compliance. Currently, only a small fraction of miners wear CPDMs, causing concerns that many miners are insufficiently protected from dust exposure. The CPDM size and weight make the device burdensome to wear and the data are not easily observable by the miner. Furthermore, CPDM data must be reported to MSHA, which discourages mine operators from using CPDMs for noncompliance purposes, such as studying dust control effectiveness.

Different types of low-cost direct-reading dust monitors have been developed to supplement the regulatory required mass-based CMDPSU and CPDM. Many of these monitors are photometers that use the principle of light scattering by an ensemble of particles to infer mass concentration [46,50–52]. As for the example shown in Figure 2c, a photometer draws particle laden air through a cyclone to achieve the desired size cut. The aerosol stream passes through a light beam, and the scattered light is measured at one or more scattering angles by photodetectors. Calibrated relationships are used to convert the scattered light intensity to particle mass concentration. Compared to CMDPSU and

CPDM, photometers have the advantages of (1) low cost, (2) lighter weight, and (3) faster time response (as low as one second). Their main disadvantage is that the relationship between light scattering intensity and particle mass concentration depends on particle refractive index, shape, size distribution, density, and relative humidity [53]. Calibrations with collocated gravimetric measurements for different mining environments are needed, which are not always feasible. Although light scattering devices are not currently used for compliance with RCMD standards, they are still useful for dust source identification, emission control technology evaluation, and alerts for excessive exposure and the need to don personal protection equipment. Owing to their low cost, portability, low power requirements, and wireless communication potential, photometers can be installed in mining microenvironments to evaluate the temporal trends and spatial distributions of dust concentrations. Their lower cost and lighter weight allows them to be used by miners that are not required to wear a CPDM, allowing more miners' exposures to be monitored [22]. Photometers provide an opportunity to further develop wearable personal dust monitors with smaller size, lighter weight, and lower cost that can be provided to every miner for non-regulatory, and possible future regulatory, exposure assessments [54,55]. For application in underground coal mines, instruments should meet safety and other permissibility requirements for potentially explosive atmospheres, and they should be rugged enough to perform in a harsh mining environment without frequent maintenance.

## 3. RCMD Particle Size Characterization

Real-time airborne particle size distribution measurements have been reviewed by McMurry [56] for atmospheric aerosols and by Giechaskiel et al. [57] for engine emissions. Methods include microscopic imaging, aerodynamic sizing, optical sizing, and electrical mobility sizing. Pros and cons of each measurement method along with the detectable particle size ranges are summarized in Table 2. Table S1 (supplemental) summarizes mining studies using these methods.

**Table 2.** A comparison of potential techniques that can be used for RCMD particle size characterization.

| Technique | Advantages | Disadvantages |
|---|---|---|
| Optical Microscopy Size range > 1μm | Visual size and morphology evaluation | Time consuming; not suitable for submicron particles; potential observational bias and errors |
| SEM Size range: ~0.01–10 μm | Morphology and size analysis; elemental characteristics; wide particle size range | Laboratory measurement; needs sample pre-preparation; slow and costly; may not be representative as a small fraction of particles are analyzed |
| Cascade Impactor Size range: ~0.01–10 μm | Wide aerodynamic diameter range; size segregated mass concentration and chemical composition; can be used for personal sampling; mechanically rugged | Ex situ analysis; long sampling duration to collect sufficient mass; particle bounce may cause bias; non-uniform deposition |
| ELPI Size range: 0.006–10 μm | In situ real-time aerodynamic size distribution; wide size and concentration ranges | Particle bouncing; blow-off from substrates; overloading of substrates; low size resolution; charging efficiency uncertainty |
| APS Size range: 0.5–20 μm | In situ real-time aerodynamic size distributions; high size resolution; easy operation | Not suitable for particles <0.5 μm; density-dependent non-Stokesian correction; liquid particle deformation and losses; low concentration limit |
| AAC Size range: 0.025–>5 μm | In situ aerodynamic size distributions; high size resolution; high transmission efficiency | Relatively slow scans (~2 min); fast rotating components; still under development/perfection |
| OPC Size range: ~0.3–10 μm | In situ real-time optical size distribution; compact and portable size; relatively low cost | Low concentration limit; dependence on particle shape and composition; non-monotonic dependence of light scattering on particle size |

**Table 2.** *Cont.*

| Technique | Advantages | Disadvantages |
|---|---|---|
| SMPS<br>Size range: ~0.003–1 μm | In situ near real-time mobility size distribution; high size resolution and accuracy for submicron particles | Relatively slow scans; not suitable for >1 μm; limitation of using radioactive neutralizers |
| EEPS/FMPS/DMS<br>Size range: 0.006–0.6 μm for EEPS and FMPS; 0.005–2.5 μm for DMS | In situ real-time mobility size distribution; high time resolution; suitable for rapidly changing aerosols | Lower size resolution than SMPS; dependence of charging efficiency on particle morphology |

SEM: scanning electron microscope; APS: Aerodynamic particle sizer; AAC: Aerodynamic aerosol classifier; ELPI: electrical low pressure impactor; OPC: optical particle counters; SMPS: scanning mobility particle sizer: EEPS: engine-exhaust particle sizer; FMPS: fast mobility particle sizer; and DMS: differential mobility spectrometer.

### 3.1. Microscopic Imaging

Microscopic analysis can determine particle size and morphology. Image processing algorithms coupled with image libraries can classify particles by their shapes and textures, identify origins, and reveal potential inhalation and health consequences [58]. A sufficient number of each particle type is required to represent exposure. Manual microscopic analysis is time consuming and requires user interpretation that may lead to observational biases and errors. Optical microscopy has been used to examine RCMD size distributions collected on filters or glass slides [59,60]. However, submicron particles cannot be determined by optical methods due to the lower size limit of ~1 μm.

Most modern RCMD applications use scanning electron microscopy (SEM) with a wider size range, from ~10 nm to tens of microns. Individual particle elemental compositions can be obtained when the SEM is equipped with an energy dispersive X-ray (EDX) detector. However, due to a lack of appropriate sampling substrates and skilled operators, RCMD has been only partially studied by this technique [16,61,62]. Computer-controlled SEM with EDX (CCSEM-EDX) reduces the personnel requirement and performs a frame-by-frame analysis for particle size, shape (e.g., aspect ratio), and elemental characteristics [16]. However, semi-volatile species evaporate under a vacuum, leading to biases for samples saturated with hydrocarbons (such as coal, organic materials, or swelling clays). "Low vacuum" and "environmental" SEMs are better suited for RCMD. Moreover, most SEMs use a fast-response solid state X-ray detector (Si(Li) detector), but render relatively low energy resolution and sensitivity for light elements (atomic number <12).

Efforts have been made to streamline SEM-EDX analysis for RCMD characterization [16,61–63]. Based on SEM imaging software, Sellaro et al. [62] used "line measurement" tools to find the long (L) and intermediate (I) particle dimensions while short (S) dimensions were estimated by assigning aspect ratios (longest divided by shortest dimension) for different minerals. The three dimensional parameters (i.e., L, I, and S) allow estimation of particle shape and volume. Based on their edge angles, particles can be classified as angular (a), transitional (t), or rounded (r) in shape, as shown in Figure 3 [64], which may be important for the particle's deposition and interaction with lung tissues. Particles are also classified into different categories based on elemental composition by EDX (e.g., quartz, carbonaceous, carbonate, etc.), each with an assumed aspect ratio and density. For easonable analysis times (i.e., ~75–90 min per sample), Sellaro et al. [62] recommended counting 100–200 particles with a magnification of 10,000× to characterize ~0.5–8 μm dimensions. This method allows a large volume of samples to be analyzed cost-effectively. The example in Figure 4 shows particle number concentration peaks at 0.5–1.0 μm followed by 1–2 μm with abundant aluminum-silica and mixed carbonaceous particles for the Roof Bolter sample.

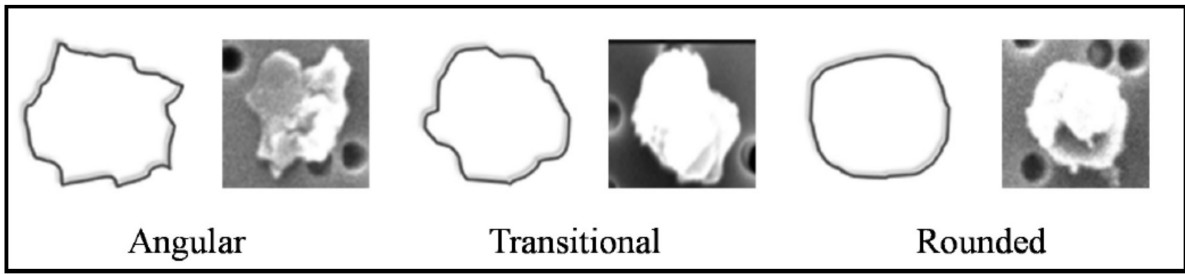

**Figure 3.** Angularity classification categories of SEM samples based on the qualitative analysis of the sharpness of particle edges [62].

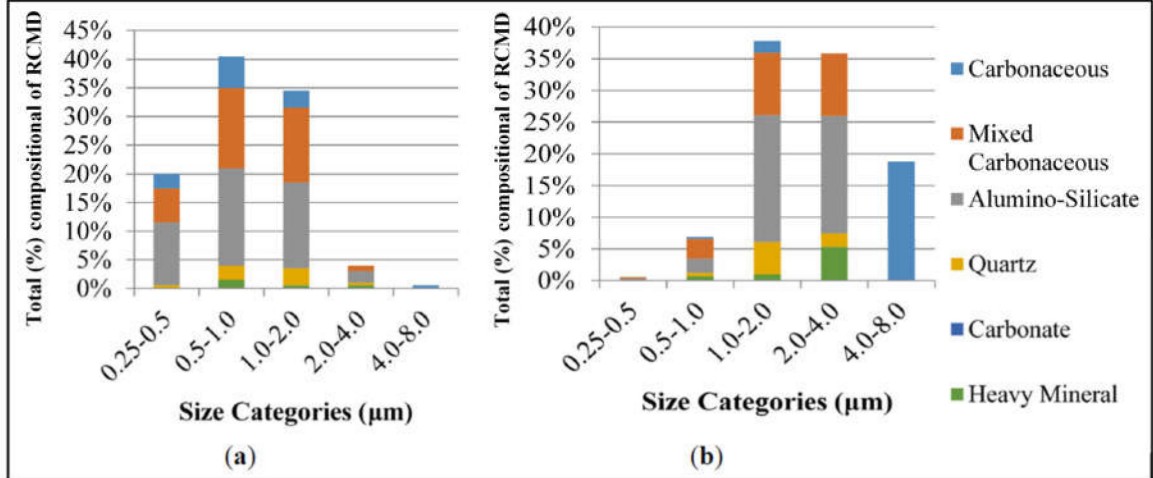

**Figure 4.** Particle size distribution by (**a**) number and (**b**) mass for the roof bolter sample (roof bolting machine); the relative number of particles in each compositional category is shown within each bar [62]. RCMD samples were collected from an underground coal mine in central Appalachia using a CMDPSU on 37 mm diameter polycarbonate filters. The center portion (9-mm diameter) of the filter was cut and attached to an SEM pin-stub for analysis.

Johann-Essex et al. [61] developed a CCSEM-EDX procedure using 1000× magnification to examine more than ~500 particles over a ~20 min sample analysis duration. However, the reduced image resolution only quantified particles ≥1 μm. For the 209 samples collected from eight underground coal mines in three Appalachian regions, particles were classified into three size bins based on their cross-sectional diameters: 0.94–2.0, 2.0–3.0, and 3.0–9.0 μm, representing small, medium, and large RCMD, respectively. Particle sizes and aspect ratios varied among geological materials, mine operating conditions, and sampled microenvironments, with more of the smaller particles in the mid-central Appalachia mines and abundant elongated particles in the south-central Appalachia mines. Higher portions of fine (i.e., 0.94–2.0 μm) and elongated particles (i.e., aspect ratios between 1.5 and 3.0) were found at production faces and in return airways. Larger particles were found at feeders and intakes (e.g., surface resuspension) with dumping and vibrations, rather than active cutting. Particles with high carbonaceous (coal) content were larger and rounder than elongated alumino-silicate particles. High quartz content corresponded to smaller particles, while high carbonate content was found in rounder particles [16].

To characterize submicron particles (0.1–1 μm), Sarver et al. [63] reanalyzed the Johann-Essex et al. [16] samples using a 20,000× magnification, finding that submicron particles dominated (>75%) the total particle number. In addition to diesel exhaust, cutting rock strata and rock dusting products were important fine particle sources. Sarver et al. [63] noted that the polycarbonate filter typically used in SEM analysis may have low collection efficiencies for submicron particles.

SEM-EDX limitations include the following: (1) the measurement is not in situ or real time; (2) particles are often collected on a polycarbonate filter and need to be transferred to an SEM stub and pretreated for analysis; (3) size-dependent particle collection efficiency and inhomogeneous deposition may lead to bias; (4) it is difficult to obtain optimal particle loadings; (5) only a fraction of collected particles are analyzed; (6) multiple magnifications are needed to analyze particles with a wide size range and it is difficult to obtain high-resolution images for particles <100 nm; and (7) the analysis is costly, time-consuming and may be subject to user interpretation.

### 3.2. Aerodynamic Particle Sizing

Aerodynamic particle sizing classifies particles based on aerodynamic diameter, which is defined as the diameter of a unit density sphere with the same settling velocity as the particle in question. The aerodynamic diameter is used to describe particle behavior in gravitational deposition, filtration, sampling, and penetration into the human respiratory system. Almost all particle-related air quality standards and sampling conventions (e.g., $PM_{2.5}$, $PM_{10}$, and respirable dust) are defined based on aerodynamic rather than geometric diameters. Four types of aerodynamic particle sizing instruments relevant to RCMD measurement are as follows: (1) cascade impactors; (2) the electrical low pressure impactor (ELPI); (3) the aerodynamic particle sizer (APS); and (4) the aerodynamic aerosol classifier (AAC).

### 3.2.1. Cascade Impactors

Inertial cascade impactors cover size ranges from a few nanometers to tens of micrometers [65]. Sequential impact stages accelerate the particle-laden flow through an array of jets positioned above flat substrates [66]. Particles with aerodynamic diameters larger than the designed cut-off size deposit on the substrate, while smaller particles follow gas streamlines moving toward the next impaction stage. The impaction nozzles are progressively smaller with each stage, thereby accelerating the particle flow to higher velocities and collecting smaller particles. Substrates, such as aluminum foils, mylar sheets, and filters, can be placed on the impaction plate for offline laboratory analysis. A filter is placed at the last stage to collect remaining particles that are too small to impact. Each substrate is weighed before and after sampling to determine mass concentrations, thereby permitting mass-based size distributions to be determined using inversion techniques that incorporate the sampling effectiveness curves for each impactor stage [67,68]. Subsequent chemical analyses of these substrates provide size-segregated chemical composition information.

Most RCMD size distributions were obtained using personal cascade impactors [24,69–71]. These small impactors with intrinsically safe sampling pumps have been worn by miners to estimate underground coal mine exposure. The Marple 290 series personal cascade impactor consists of up to eight impaction stages and a backup filter, with 50% cut-off diameters of 21.3, 14.8, 9.8, 6.0, 3.5, 1.55, 0.93, and 0.52 µm at a flow rate of 2 L/min [72]. The Sioutas personal cascade impactor consists of four stages with diameters of 2.5, 1, 0.5, and 0.25 µm at a flow rate of 9 L/min [73]. Chen et al. [74] developed a 10 impaction stage personal impactor collecting 0.06–9.6 µm particles at a flow rate of 2 L/min. Due to the low flow rate of personal cascade impactors (typically 2 L/min), a long sampling time is needed to collect sufficient mass for reliable gravimetric and chemical analyses. A 10 stage micro-orifice uniform deposit impactor (MOUDI) with 50% cut-point diameters of 10 to 0.056 µm at 30 L/min, or the 13 stage MOUDI with cut-points of 10 to 0.010 µm at flow rates of 10 or 30 L/min have been used to reduce sample durations, increase collected mass, and improve sizing resolution [75]. Figure 5 shows mass-based size distributions measured in a diesel-powered coal mine having more submicron particles than in an entirely electric-powered coal mine, indicating the large contributions of diesel engine emissions to submicron particles [76,77]. An intrinsically safe pump is required to operate the MOUDI in underground coal mines.

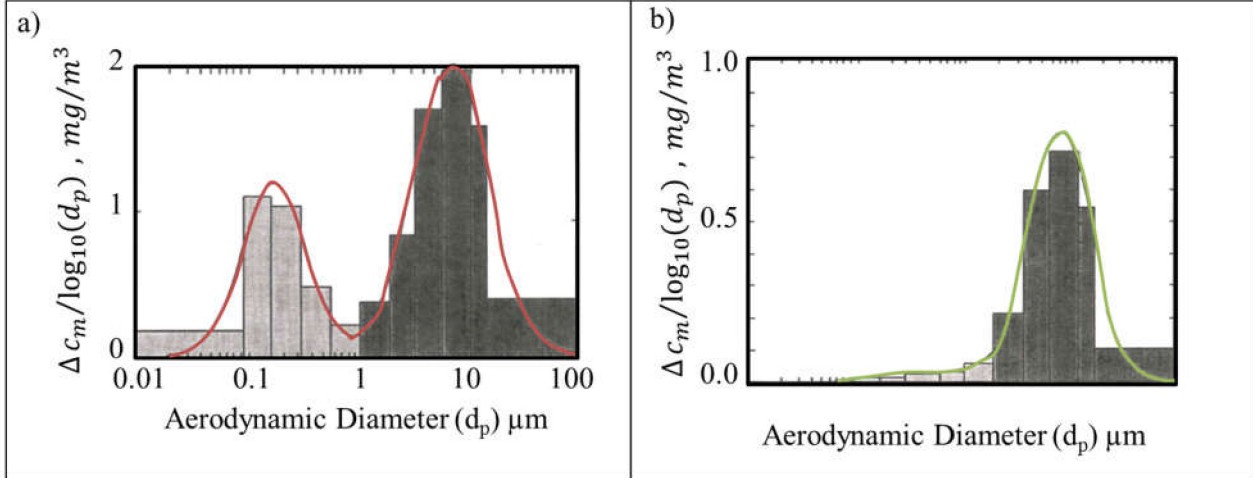

**Figure 5.** Mass-based size distribution measured by MOUDI at (**a**) a coal mine with diesel equipment, and (**b**) a coal mine with all electric equipment [76,77].

Particle bounce, wherein a larger deposited particle is re-entrained into the airflow for deposition on a subsequent stage, is a cause of uncertainty that shifts the distribution to smaller sizes compared to that in the atmosphere. Particles are not uniformly deposited across the impaction surface (except for the rotating MOUDI), with patterns reflecting the nozzle shapes and positions. As a result, filters cannot be sectioned for submission to different chemical analysis methods that assume a uniform deposition.

The quartz crystal microbalance coupled with a MOUDI (QCM-MOUDI) [78] determines real-time mass concentrations from the vibration frequency change of the quartz crystals. The QCM-MOUDI (model 140, TSI Inc., Shoreview, MN, USA) consists of a $PM_{2.5}$ inlet and 6 QCM stages (2.44, 0.96, 0.51, 0.305, 0.156, 0.074, and 0.045 μm). With a flow rate of 10 L/min, it records a mass-based size distribution every second, although longer integration periods are more accurate. Particle bounce is reduced by controlling the inlet flow relative humidity (RH) in the range of 40–65%. However, frequent cleaning is needed to prevent dust overloading and bounce in a harsh mining environment.

### 3.2.2. Electrical Low Pressure Impactor (ELPI)

The ELPI (Dekati Ltd., Kangasala, Finland) measures particle size distribution as a function of aerodynamic diameter with high time resolution (as fast as every 0.1 s). Particles are sampled through a unipolar corona charger that imposes a well-defined charge distribution on particles independent of their initial charging state. The charged particles then pass through a low-pressure cascade impactor with electrically isolated collection stages. Particles are collected on impaction substrates depending on their aerodynamic diameters, and the electric charges carried by particles are measured by a multichannel electrometer [79]. Data inversion algorithms are used to convert raw current readings to particle number concentrations [80–82]. The ELPI consists of 14 impaction stages and 1 backup filter for particle sizes of 6 nm to 10 μm. The main advantage of the ELPI for RCMD measurements is that it covers a wide size range with fast response times to distinguish rapidly changing nano- (e.g., DPM) and supermicron-particles. The aerodynamic diameter-based number distribution can be converted to mass distribution with less uncertainty due to variable particle density and shape. However, knowledge or assumptions of the effective density are required to reconcile differences between mobility diameter-dependent charging efficiency and aerodynamic diameter-dependent impaction separation [83]. ELPIs have been used in engine emission testing with good agreement between the ELPI and gravimetric mass concentrations [84–87].

ELPIs have the same shortcomings as cascade impactors, such as particle bounce and blow-off from substrates and a relatively low size resolution. Particle charging efficiency

depends on particle morphology, concentration, carrier gas composition, and relative humidity [88,89]. The aging and contamination of the charger can cause additional errors [90,91]. For RCMD measurement in underground coal mines, overloading of impaction stages, frequent cleaning, and intrinsic safety are the main limitations.

Bugarski et al. [92] used an ELPI to evaluate the effects of the longwall moving process in an underground trona mine (at an isolated zone test site) and found aerosol size distributions having two, three, or even four lognormal modes. As expected, diesel engines emitted submicron particles that dominated number concentrations, while supermicron dust particles dominated mass concentrations. Diesel-powered engines were found to be the primary source of both submicron aerosols and resuspended coarse dust. Although electrically powered vehicles did not directly generate tailpipe emissions, they also contributed to resuspended dust.

### 3.2.3. Aerodynamic Particle Sizer (APS)

When particles are rapidly accelerated through a nozzle, they attain different velocities depending on their inertia, which is a function of particle size and density [93]. Smaller particles are accelerated faster due to their lower inertia. The APS measures the time-of-flight for particles passing through a path bounded by two laser beams of known separation downstream of an accelerating nozzle to infer velocities. The times-of-flight are then converted to aerodynamic diameters based on a calibration curve [94]. The current APS (model 3321, TSI Inc., Shoreview, MN, USA) measures 52 size bins every second for aerodynamic diameters of 0.5–20 μm with simultaneous optical sizing for 0.37–20 μm diameter particles over a concentration range of ~0.001–1000 particles/$cm^3$.

APS measurements have several limitations [95,96]. For larger particles, the acceleration velocity depends not only on aerodynamic diameter, but also on gas density, gas viscosity, particle density, and particle shape [97–99]. Corrections are possible when these properties are known; otherwise, the reported aerodynamic diameter may be biased [99]. The size of a particle with a density of 0.8 g/$cm^3$ can be underestimated by as much as 5%, and a particle with a density of 2 g/$cm^3$ can be overestimated by as much as 10%. Liquid droplets may deform during acceleration, leading to size underestimation. The degree of distortion depends on liquid viscosity and surface tension [100,101]. Liquid particles also have higher transport losses than solid particles [102]. The APS has a relatively low concentration limit (1000 particle/$cm^3$) before coincidence errors (multiple particles passing through the laser beams) become significant. A dilutor can be used to reduce the particle number, but particle losses in the dilutor could lead to uncertainties for concentrations in the larger size channels.

The APS has been used in laboratory studies to measure the sampling effectiveness curves of aerodynamic classification devices such as cyclones [31,37,40,103] and impactors [73,75] that are used in mine applications. It has also been used for ambient aerosol size distributions, including locations close to mining areas [104–107]. Concurrent mobility and aerodynamic size distribution measurements or using the APS to measure mobility classified particles allow for the estimation of particle effective densities and dynamic shape factors [108–110]. Due to the low concentration range, the APS has had limited use for in-mine size distributions. Saarikoski et al. [111] used two scanning mobility particle sizers (SMPS) and an APS to measure size distributions in the range of 2.5 nm–10 μm in an underground chrome mine. It was found that submicron particles from diesel engine exhaust and explosion combustion products yielded higher numbers and mass concentrations than mechanically generated coarse particles.

### 3.2.4. Aerodynamic Aerosol Classifier (AAC)

The AAC (Cambustion Ltd., Cambridge, UK) classifies particles based on a balance between centrifugal and drag forces [112,113]. The AAC consists of two concentric cylinders that spin at the same speed forming an annular classifying region. Aerosols enter the AAC near the wall of the inner cylinder, and traverse through clean sheath air towards the

outer cylinder due to the centrifugal force. Particles with different inertias are separated into different trajectories: larger particles adhere to the outer cylinder wall; smaller particles exit with the excess sheath flow; and particles with the selected size exit through the monodisperse aerosol sampling port. By rotating the cylinders at different speeds, different particle sizes can be selected and measured by a condensation particle counter (CPC), generating number size distributions based on aerodynamic diameter. The AAC (1) covers a wide size range from 0.025 to >5 μm; (2) does not require particle charging, resulting in transmission efficiencies 2–5 times higher than the SMPS that has an electrostatic operating principle; and (3) can be combined with mobility classification or mobility size distribution measurements to quantify particle effective densities, dynamic shape factors, and charge states [114]. Its drawbacks are that it takes several minutes to scan a size distribution, and the rotating components pose reliability challenges, particularly in harsh mining environments. The AAC is a relatively new instrument and its design is still being perfected for real-world size distribution measurements.

### 3.3. Optical Particle Sizing

Single particle optical particle counters (OPC) or spectrometers (OPS) measure particle sizes based on the amount of light scattered by individual particles, in contrast to photometers that measure total light scattering from an ensemble of particles [56,115,116]. In an OPC, the light beam and particle stream are designed to reduce the probability of multiple particles being present in the sensing volume at the same time. The scattered light is converted to a proportional electrical pulse by a photodetector. The height or area of the pulse is used to infer particle diameter based on a predefined calibration curve, typically generated using spherical particles of known sizes and composition (e.g., polystyrene latex spheres). OPC designs differ in light sources (e.g., white light or wavelength-specific laser), scattered light collection angles (e.g., forward or side scattering), and photodetectors (e.g., photodiode or photomultiplier tube). Due to scattering by air molecules and electronic noise in the circuitry, most OPCs measure particle sizes in the range of ~0.3–10 μm. Advanced instruments can detect diameters as low as 0.05 μm [117]. Similar to photometers, the main advantages of OPCs are (1) fast time response (~1 s), (2) compact and portable size, and (3) relatively low cost. However, OPCs have low concentration limits (typically several thousand particles per cubic centimeter). Coincidence errors lead to inaccurate size and concentration measurements [118,119]. This problem can be partially overcome by combining single particle counting with photometry as implemented in the DustTrak DRX (TSI Inc., Shoreview, MN, USA), which measures size-segregated concentrations up to 400 mg/m$^3$ [53,120]. The light scattering signal depends not only on particle size, but also on particle refractive index and shape. Therefore, OPCs report "optical equivalent size" based on the particles used to establish the calibration curve, which may deviate from a particle's geometric, aerodynamic, or mobility size. The light scattering intensity vs. particle size curve is often not monotonic, especially for particles larger than 1 μm, leading to lower sizing resolutions and higher uncertainties [56].

OPC applications in mines have been limited, owing to the diverse refractive indices and non-spherical shapes of coal dust. Liu et al. [121] calibrated a near-forward scattering OPC by aerosolizing a small quantity of finely ground coal dust with a fluidized bed. A differential mobility analyzer (DMA) selected monodisperse particles over a size range from 0.4 to 2.4 μm. The experiments found that coal particles of the same mobility sizes generated lower OPC responses than transparent oil particles owing to the light-absorbing nature of the coal. The pulse height distributions from monodisperse coal particles were also broader than those for oil particles, likely due to their irregular shape. Without proper OPC calibration, coal dust sizes can be underestimated by up to fivefold [121,122]. However, OPC systematic sizing errors can be minimized by calibration with representative coal dusts. Barone et al. [123] applied ray tracing with diffraction on facets and T-matrix theories to adjust the responses of an OPC for submicron and micron size coal particles, respectively. This method accounted for the refractive index and non-spherical shape when computing

coal dust diameters from light scattering theory. The size distributions measured by the OPC had reasonable agreement with APS, CCSEM, as well as cyclone-separated and sieve-segregated sizes. Marple and Rubow [124,125] calibrated an OPC to report aerodynamic sizes by measuring the sampling effectiveness curves of an impactor inlet and comparing to its known aerodynamic size penetration curve.

### 3.4. Electrical Mobility Particle Sizing

Electrical mobility is the most widely used technique to measure size distributions in the submicron size range [126]. There are two major designs: voltage scanning and non-scanning. The scanning mobility particle sizer (SMPS) consists of a bipolar charger, a differential mobility analyzer (DMA), and a condensation particle counter (CPC) [127]. Particles are first passed through a bipolar charger to obtain a well-defined stationary charge distribution [128,129]. The charger often uses a small quantity of radioactive material (e.g., krypton-85, polonium-210, or americium-241) or soft X-rays to ionize molecules in the air, which subsequently attach to particles by diffusion charging [88,130–133]. Particles then enter the DMA, where charged particles are separated into different trajectories by an electric field, depending on their electrical mobility [134]. At a given voltage, only particles of a given mobility size pass through the DMA to be counted by a CPC [135]. By varying the DMA voltage, particles with different sizes are selected and counted. The data inversion algorithm generates size distributions by taking into account the charge distribution, DMA transfer function, CPC counting efficiency, time constants, and particle transport losses [127,136,137]. Depending on DMA and CPC designs, the SMPS can quantify size distributions from several nanometers to several hundred nanometers every 1–2 min [138]. Recent advances in DMA, CPC, electrometers, and inversion algorithms include the following: (1) measuring particles down to 1 nm [139–146], (2) measuring size distributions in less than one minute [137,147–150], and (3) more portable and rugged designs [151,152].

As the SMPS measures submicron particle size distributions, it is often used in parallel with an APS [110,116,153,154] or an OPC [155–158] to cover a wider size range. Skubacz et al. [159] used the SMPS and APS to measure 0.015–0.698 μm and 0.5–20 μm particle size distributions, respectively, in an underground coal mine. They observed high concentrations of ultrafine particles when electric-powered mining machines were in operation. Saarikoski et al. [160] combined an SMPS, OPC, and ELPI for particle size and a soot particle aerosol mass spectrometer (SP-AMS) for particle chemical composition. They found that engine exhaust emissions (dominated by organic matter and black carbon) accounted for 35–84% of submicron particle mass, and blasting (dominated by organic matter, sulfate, nitrate, ammonium, and black carbon) produced 7–60% of submicron particles' mass in an underground chrome mine.

In contrast to the SMPS, which scans voltage to obtain size distributions, a non-scanning mobility spectrometer uses multiple detectors to measure mobility-separated particle concentrations. Commercially available non-scanning instruments include the engine exhaust particle sizer spectrometer (EEPS; model 3090, TSI Inc., Shoreview, MN, USA), fast mobility particle sizer spectrometer (FMPS; model 3091, TSI Inc., Shoreview, MN, USA), and differential mobility spectrometer (DMS; Cambustion Ltd., Cambridge, UK). To increase detector signals, these instruments use unipolar chargers to charge the incoming particles, separate particles based on electrical mobility, and measure size-segregated particle concentrations using a series of electrometers [89,161–166]. These instruments can produce size distribution data as fast as every 0.1 s, and therefore are suitable for studying fast changing aerosols, such as in transient engine exhaust measurements. Their main disadvantages include (1) lower size resolution than the SMPS, (2) larger uncertainties in the unipolar charge distribution due to dependence on particle morphology [89,164], and (3) lower concentration sensitivity due to electrometer measurements.

Several studies have applied electrical mobility particle sizers in mining environments, particularly those related to engine exhaust. Bugarski and Hummer [167] used a FMPS to measure diesel-powered vehicle emissions in an underground mine to assess relative

contributions of different types and categories of diesel engines to submicron aerosols and to assess the effectiveness of diesel emission control technologies. They found that replacing a U.S. EPA pre-tier engine with a Tier 3 engine resulted in 41% lower particle number concentrations. A retrofitted disposable filter element reduced particle number emissions by 77–92%, while a retrofit sintered metal filter reduced particle emissions by 93%. Bugarski et al. [168] also used an FMPS and an ELPI to measure size distirbutions emitted by a diesel-powered personnel carrier vehicle and by a manual metal arc welder in an underground mine. The FMPS size distributions for both diesel exhast and welding fumes had modes at ~10 nm and ~70 nm, with welding aerosols having an additional mode at ~140 nm. The ELPI data demonstrated that neither diesel exhaust nor welding generated micron-sized particles.

### 3.5. Evaluation for Size Distribution Measurements in Mines

The ELPI has several advantages for RCMD size distribution measurements. First, it measures a wide aerodynamic diameter range from 6 nm to 10 μm, covering both diesel exhaust and mechanically generated dust particles. Second, it has a high time resolution (0.1 s), allowing it to measure size distributions of fast changing aerosols. Third, because the ELPI measures aerodynamic diameter, the conversion from number distribution to mass distribution has less influence from particle properties than SMPS or OPCs, making the integrated mass concentrations more comparable to the regulatory required gravimetric mass concentrations. A recently developed ELPI algorithm reports high resolution (up to 500 size channels) inverted size distributions [82]. The full size distribution allows for calculations of size-segregated particle surface and mass concentrations, permitting evaluation of the effects of these alternative metrics on RCMD health effects. Different substrates can be used in the ELPI to collect particles for microscopic and chemical analyses. In addition to ELPI, cascade impactors and SEM-EDX analysis can complement RCMD characterization. Cascade impactors do not only derive mass distributions, they also allow analysis of particle chemical compositions in different size ranges. SEM-EDX analysis provides additional information about particle morphology and particle-level chemical composition.

## 4. Chemical Composition of RCMD

The complex and heterogeneous nature of underground RCMD can include over 50 different elements and their oxides [169,170], consisting of 40–95% coal and 5–60% mineral mixtures [171]. Table 3 summarizes commonly found inorganic minerals in RCMD including carbonates, silicates, and sulfides/sulfates [172,173].

Several research projects have addressed the effect of silica as an occupational health hazard in coal mines [1,2]. However, recent epidemiological studies suggest that other elements such as bioavailable Fe in minerals may contribute to CWP [174,175]. RCMD minerals partially originate from damaged surrounding geological formations on the walls, roof, and floor. There are additional sources in coal mining that are not directly related to coal production, such as resuspended dusts from support system installations and low-silica limestone dust that is added to prevent explosions. Diesel-powered underground equipment can also generate large amounts of fine particles. In the USA several rules are promulgated to limit DPM exposure in metal and nonmetal underground mines [176,177].

At present, crystalline silica (quartz) is the main component specified in mine regulations owing to its adverse health effects. Quartz abundances in coal dust vary by tenfold, with 4.2 to 14% in Belgian mines (prior to 1959) [178], 2.4 to 5% in German mines (1971) [179], 1.5 to 10.3% in British mines (1970–1975), and 2.5 to 7% in U.S. mines (1985–1992) [180]. Figure 6 shows the average percentage of quartz found in RCMD samples from eight different mines [61]. It is apparent that the percentage of quartz differs among mining operations and locations. Generalizations about RCMD composition cannot be made without additional characterization of coal dust mineralogy. The silica mass percentage in RCMD also differs between different surface and underground coal mines [181]. From 1997 through 2011, surface mines generated much higher silica as compared to underground

coal mines. Cauda et al. [181] found that the 95th percentiles of silica abundance were 19.2% and 36.6% in underground and surface mine dust samples, respectively.

**Table 3.** Typical RCMD minerals [172,173].

| Classification | Mineral | Formula |
|---|---|---|
| Carbonates | Siderite | $FeCO_3$ |
| | Dolomite | $CaMg(CO_3)_2$ |
| | Ankerite | $Ca(Fe, Mg, Mn)(CO_3)_2$ |
| | Calcite | $CaCO_3$ |
| | Magnesite | $MgCO_3$ |
| Silicates | Illite | $K_{0.65}(Al,Fe,Mg)_{2.0}[Al_{0.65},Si_{3.5}]O_{10}(OH)_2$ |
| | Kaolinite | $Al_2[Si_2O_5](OH)_4$ |
| | Sericite | $KAl_2(AlSi_3O_{10})(OH)_2$ |
| | Smectite | $M_x(Si_4)(Al_{2-x},((Mg,Fe^{3+})_x)O_{10}(OH)_2.nH_2O$ |
| | Quartz | $SiO_2$ |
| | Montmorillinite | $M_x(Si_4)(Al_{2-x},((Mg,Fe^{3+})_x)O_{10}(OH)_2.nH_2O$ |
| Sulfides/Sulfates | Marcasite | $FeS_2$ |
| | Pyrite | $FeS_2$ |
| | Melnikovite | $FeS_2$ |
| | Sphalerite | $ZnS$ |
| | Galena | $PbS$ |
| | Chalcopyrite | $CuFeS_2$ |
| | Gypsum | $CaSO_4.2H_2O$ |
| | Jarosite | $KFe_3(SO_4)_2(OH)_6$ |
| | Hydrated iron sulfates | $FeSO_4 \cdot xH_2O$ |

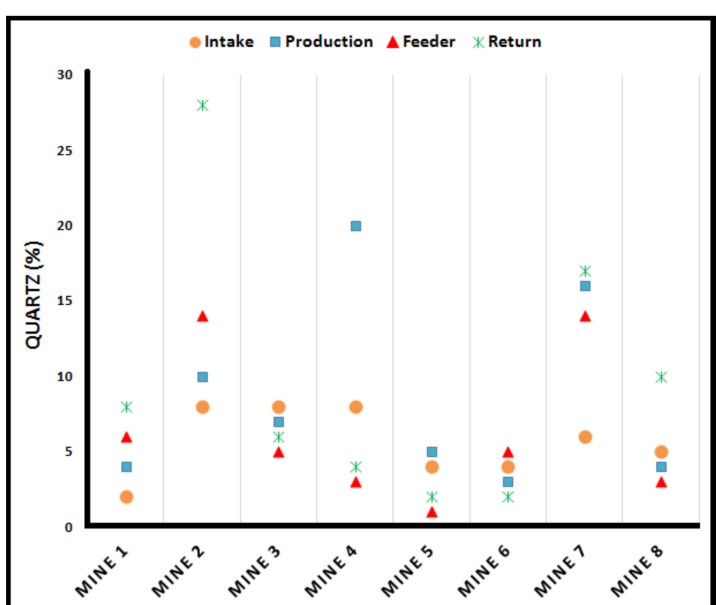

**Figure 6.** Average quartz distributions (% of characterized particles) with different locations/operations in mid-central Appalachia (Mines #1–4), northern Appalachia (Mines #5–6), and south-central Appalachia (Mines #7–8) [61].

As easily available (near surface and thick seam) high quality coal resources are exhausted, thinner coal seams are being exploited, resulting in more non-coal dust from the walls, immediate roof, and floors [182–184]. Sandstone is one of the common layers surrounding coal seams that contains crystalline silica [185]. More powerful machinery also increases finer particle silica in the generated dust [186] from different parts of an underground mine (e.g., production shafts, adits, drifts, stopes, etc.). Additionally, different rock types are found to affect the dust mineralogy [16,61,62].

This section examines the use of thermal (i.e., differential thermal analysis, thermogravimetric analysis, thermal/optical analysis, and thermal desorption) and spectroscopic (i.e., light transmission/absorption, energy dispersive X-ray, X-ray fluorescence, inductively coupled plasma-mass spectrometry, X-ray diffraction, Fourier-transform infrared spectroscopy, Raman spectroscopy, and [13]C and [1]H nuclear magnetic resonance spectroscopy) analysis methods in RCMD chemical analysis. Table S2 (Supplementary Material) presents a summary of some major studies related to RCMD's chemical analysis.

## 4.1. Thermal Analysis Methods

Thermal methods determine chemical composition by decomposing samples under controlled atmospheric and heating conditions and relating changes to melting, vaporization, and combustion temperatures of target compounds. Thermal analysis techniques are used to (1) identify the utilization and ranking of coal (e.g., degree of organic metamorphism or coalification) [187], (2) determine phase transitions and chemical reaction rates [188] for investigating coal dust mineralogy [189–191], and (3) identify organic materials [192–195]. Warne [172] found 87 minerals along with products of chemical reactions in coal by thermal analysis. Thermal analysis preheats the sample to remove moisture (first stage), evaporates volatile materials (second stage), and combusts the remainder (third stage) [187]. Klaja et al. [188] estimated mineral water content in two different temperature steps, including evaporation of adsorbed water (40–150 °C) and structural water (440–620 °C). Few studies have used this method for RCMD characterizations.

Thermal methods do not distinguish among complex mine dust compositions with similar thermal properties [196]. However, their outputs can complement information gained from spectroscopic techniques to identify potentially toxic organic components. Five thermal methods are summarized below, including differential thermal analysis (DTA), differential scanning calorimetry (DSC), thermogravimetric analysis (TGA), thermal/optical analysis by reflectance and transmittance (TOT/TOR), and thermal desorption (TD).

### 4.1.1. Differential Thermal Analysis (DTA)

For DTA, the target sample is heated under controlled conditions to compare the sample with reference temperatures according to the exothermicity or endothermicity of chemical reactions during heating. The resulting peaks are compared with a standard database to identify dust composition [197,198]. When applying this method to identify minerals in coal samples, a high concentration of clays (>50%) might inhibit the characterization of silica, carbonates, and sulfides. Under an inert atmosphere, DTA exhibits minor peaks, which allows for comparison with standard patterns to identify mineral matter [199–201]. Table 4 shows detection limits for DTA under an inert nitrogen ($N_2$) atmosphere as reported by Warne [202]. Similar curve features can be observed when substituting $CO_2$ for $N_2$ to inhibit coal combustion, resulting in well-defined mineral peaks, except for carbonates [199].

**Table 4.** Detection limits of minerals in an inert nitrogen (N2) atmosphere by DTA [199,202].

| Mineral Matter | Detection Limit (wt %) |
| --- | --- |
| Pyrite and marcasite | 0.5% |
| Calcite, magnesite, dolomite, and ankerite | 1% |
| Siderite and kaolinite | 2% |
| Quartz | 2 to 5% |
| Montmorillonite | 15% |
| Illite | Up to 30% |

### 4.1.2. Differential Scanning Calorimetry (DSC)

DSC is based on the heat needed or released during a phase change. The energy needed to keep the temperature constant between a substance and an inert reference material under identical cooling and heating rates is measured [203]. DSC complements spec-

troscopic methods for determining organic and inorganic compounds in RCMD [204–206]. Klaja et al. [188] utilized this method to identify organic materials in coal samples with results comparable to those of the pyrolysis method [195]. High temperature DSC has been calibrated to measure the decomposition enthalpies of minerals in coal samples [207–209].

### 4.1.3. Thermogravimetric Analysis (TGA)

For TGA, a small sample of coal dust is heated in a selected atmosphere with a temperature gradient and the weight loss by sensitive gravimetry is recorded at each temperature interval. Pyrolysis behavior is analyzed to determine the reactivity and kinetic parameters of sample combustion [210,211]. Owing to its short analysis time and good precision, TGA can be used for evaluating coal chemical and physical properties of many samples [212]. Three mass fractions in RCMD, i.e., coal, non-carbonate minerals, and carbonate content can be determined [213,214]. TGA can quantify calcite [215], and can be integrated with other standard methods to identify silica content and total mass [215,216].

### 4.1.4. Thermal/Optical Analysis by Reflectance and Transmittance (TOR/TOT)

This method is used for combustion source (e.g., DPM) emissions that contain organic carbon (OC) and elemental carbon (EC) [203]. Watson et al. [217] summarize twenty methods to separate OC and EC, which include heating the sample to different temperatures, oxidizing the evolved materials to $CO_2$, quantifying evolved $CO_2$ with an infrared absorption detector, or reducing to methane ($CH_4$) for detection by a flame ionization detector (FID). The NIOSH Method 5040 used TOT to determine EC as an indicator for DPM [218–220]. The IMPROVE_A protocol developed by Chow et al. [221] for U.S. long-term $PM_{2.5}$ speciation networks classifies carbonaceous aerosol into four OC fractions (OC1–OC4) in an inert 100% helium (He) atmosphere and three EC fractions (EC1-EC3) in a 98% He/2% $O_2$ atmosphere [221–225]. Recently, reflectance and transmittance have been measured at multiple wavelengths from 405 nm to 980 nm, allowing for the estimation of brown carbon (BrC) and black carbon (BC) content in particles [226–229]. This technique offers a useful identification method for integrated filter samples [230] that can be effectively applied to analyze DPM and RCMD filter samples. Additionally, the thermal carbon fractions are useful for exploring RCMD properties for source apportionment receptor modeling.

### 4.1.5. Thermal Desorption (TD)

TD involves collecting released compounds from a heated sample into a gas carrier from which the evolved material is concentrated for submission to gas-chromatographic analysis. The separated organic compounds are detected by a flame ionization detector (FID) or, more specifically, by a mass spectrometer (MS). No sample pretreatment is required, making TD more cost-effective than organic speciation by solvent extraction [231–233]. Variable solvent extraction efficiencies and potential sample contamination limit accurate quantification of organic compounds [234–237]. There are thousands of organic compounds that are potentially harmful to human health. Watson et al. [238] list functional groups and some of the specific compounds that can be practically measured with thermal desorption-gas chromatography-mass spectrometry (TD-GC-MS) [239]. Providing comparable results to those of solvent extraction, TD-GC-MS can be applied to analyze nonpolar organic compounds in coal mine dust qualitatively and quantitatively [240].

### 4.2. Spectroscopic Analysis for Chemical Compounds

Spectroscopic techniques determine chemical compositions by multiwavelength radiation interactions with matter and by different forms of chromatography. These methods produce spectra (response as a function of wavelength or time) that are indicative of the elements or compounds, with their intensity related to concentration levels. Several of these methods are non-destructive and are performed on a small portion of a filter sample, thereby increasing the types of analyses that can be applied. The following section surveys

several spectroscopic techniques that are applicable to analyze mineral dust, including EDX (energy dispersive X-ray), XRF (X-ray fluorescence), ICP-MS (inductively coupled plasma-mass spectrometry), XRD (X-ray diffraction), FTIR (Fourier-transform infrared spectroscopy), Raman spectroscopy, and NMR (nuclear magnetic resonance).

### 4.2.1. Energy Dispersive X-ray (EDX)

EDX directs X-rays, electrons, or protons that shift inner shell electrons to higher energy levels. Each element emits characteristic X-rays as the electron transitions back to its ground state, allowing the elements to be identified and quantified [62]. The filter substrate also scatters the incident X-rays, and must be subtracted by analyzing blank filters [241]. EDX can be applied to individual particles in SEM and to bulk samples with a laboratory analyzer.

EDX used in SEM can be labor intensive. Recent efforts have attempted to streamline the processes for characterizing RCMD samples from different mine regions and microenvironments [16,61,242]. Using a spreadsheet program to automate the computational analysis, Sellaro et al. [62] increased the number of particles analyzed by five- to tenfold. Particles in RCMD samples were classified as carbonaceous, mixed carbonaceous, aluminosilicate, quartz, carbonate, or heavy minerals that include twelve elements (i.e., carbon, oxygen, sodium, magnesium, aluminum, silicon, sulfur, potassium, calcium, titanium, iron, and copper). Unclassified particles are grouped as "other".

Applying a computer controlled EDX (CCEDX) technique, Johann-Essex et al. [16] increased the data acquisition rate by 25-fold for carbonaceous, alumino-silicate, quartz, carbonate, and heavy minerals. These studies, similar to previous studies by EDX [16,62,241], have captured only a small fraction of particles, and the elemental concentrations were not quantified.

### 4.2.2. X-Ray Fluorescence (XRF)

XRF is EDX applied to the thin layer of particles collected on the surface of a filter, typically Teflon or polycarbonate membranes [243]. XRF is automated, non-destructive, and more efficient than other multi-elemental analyses such as atomic absorption spectroscopy, which require an acid extraction followed by single element quantifications [203]. Photon radiation with 1000–30000 eV generated by an X-ray tube is applied to the sample. As in EDX, this energy ejects the lower shell electron to a higher level with a characteristic X-ray photon emitted when it returns to its ground state [244]. The X-ray peaks are related to elemental concentrations by comparison to thin film standards with appropriate adjustments for overlapping peaks, matrix interferences, and absorption of the particles and filter [240]. Heavy metal concentrations in mine dust samples from different environments have been compared using XRF [245]. XRF requires minimal sample handling and can identify >50 elements from sodium to uranium with low detection limits [240]. RCMD samples collected by a CMDPSU or laboratory resuspended bulk material from mines are most suitable for XRF analysis. However, XRF is not sensitive to the low concentrations of several rare-earth elements (lanthanide series) or light elements (e.g., Li, Be, and B) [246].

### 4.2.3. Inductively Coupled Plasma-Mass Spectrometry (ICP-MS)

ICP-MS complements XRF by providing lower detection limits for elements including rare-earth elements and isotopes. There is an increasing desire to identify multiple elements by high-sensitivity ICP-MS in health studies [247–252]. Disadvantages of ICP-MS include (1) destructive filter extraction in strong acids, (2) potential sample contamination during sample extraction, (3) incomplete extraction efficiencies, and (4) being labor intensive and associated with high cost when compared to XRF [246].

Acid-digested sample extracts are ionized in a plasma torch and passed through a quadrupole MS, which sorts elements based on their mass-to-charge ratios. Laser ablation ICP-MS uses a high-powered pulsed laser to vaporize a portion of the filter for direct injection into the ICP-MS [253,254], but it has not proven to be as quantitative as the acid

extraction. Calibration standards, equivalence testing, and optimization are needed to establish reproducibility, standardization, and detection limits [240].

For the 74 RCMD samples collected from eight mines in central and northern Appalachia, Sarver et al. [63] found K, Si, Mg, Al, Fe, and Zn in ~80% of the acid-digested samples, ranging from 10–200 mg/g. Trace elements Cu, Ba, Co, Ni, Mn, Cr, and Ag were found in ~30% of the samples, and <15% of the samples contained detectable amounts of Sr, As, Pb, V, and U. Se, Cd, and Sn concentrations were below the minimum detectable limits [63].

### 4.2.4. X-ray Diffraction (XRD)

XRD determines the composition and structure of crystalline substances in solid dust and rocks, qualitatively and semi-quantitatively [255–258]. XRD analysis irradiates the sample with X-rays to generate a specific diffraction pattern determined by a rotating crystal that provides crystal structure, molecular configuration, and composition [203]. Coal and coal dust were found to contain high background intensity of the diffractograms by XRD [254,259]. The diffraction direction is associated with diffraction intensity due to molecular arrangements in the unit cell as well as the size and shape of crystalline cells. The crystalline structure of carbonaceous material is detected [260]. Hirsch [261] and Diamond [262] provide a statistical interpretation of XRD profiles for carbonaceous materials with low-crystallinity that reveal the capability of XRD to measure the non-crystalline portion of the coal.

XRD results for coal structural parameters can be evaluated by the Scherrer equation and Gaussian curve fitting method [260]. The Bragg formula reveals that the position of a diffraction peak (Bragg angle 2θ) is a function of interplanar spacing, indicating that peak fluctuations represent interplanar spacing. XRD analyses are commonly applied to indicate mineral composition of the coal and the interaction of mineral matter during conversion processes [203].

Warren [263] initiated XRD analysis of carbonaceous materials using a set of equations that relate peaks to material type, thereby separating four types of carbon black [263,264]. The computer program (SIROQUANT) established by CSIRO Australia shows the capability of identifying and quantifying several minerals [265,266]. Lu et al. [267] also determined the ultrafine structure of coals based on the X-ray intensity profile in the medium and high range of scattering angle. Figure 7 shows the differences in microcrystalline structures and mineral phases of coal dust before and after an explosion. XRD analysis identified quartz, calcite, kaolinite, and dolomite, which are often present in clay and saline minerals surrounding the coal layer [257]. Primary elements in the original coal dust were C, O, Al, Ca, Si, Mg, and Fe [268]. Based on this ability, the NIOSH standard method (NIOSH Manual of Analytical Method 7500) [269] suggested using XRD to quantify silica in dust particles on a filter. However, using XRD analysis for coal samples has some limitations/difficulties because of (1) bulk sample preparation for obtaining representative and homogeneous analysis, (2) particle size distributions of the analysis samples, (3) qualitative matrix composition of the analysis sample, and (4) interferences of spurious phases with the critical analyses [270].

Diffractograms in the diffraction angle 2θ region of 15−35° have been applied to determine the organic part of coal dust. The broad hump (Figure 8) can be fitted to two Gaussian peaks around 20° and 26° representing the γ-band and the Π-band (002), respectively. The γ-band represents the packing distance of saturated structures, and the Π-band indicates the spacing of aromatic ring layers [271]. The position, intensity, full width at half-maximum, and the integrated area of these fitting peaks can be determined. The number of aromatic and aliphatic carbon atoms can be approximated by the areas under the γ-band and Π-band, respectively [267,272]. The peak intensities (I) at the γ-band and Π-band positions are used to classify coal ranks using the following equation [260]:

$$\text{Coal rank} = I_{\Pi}/I_{\gamma} \tag{1}$$

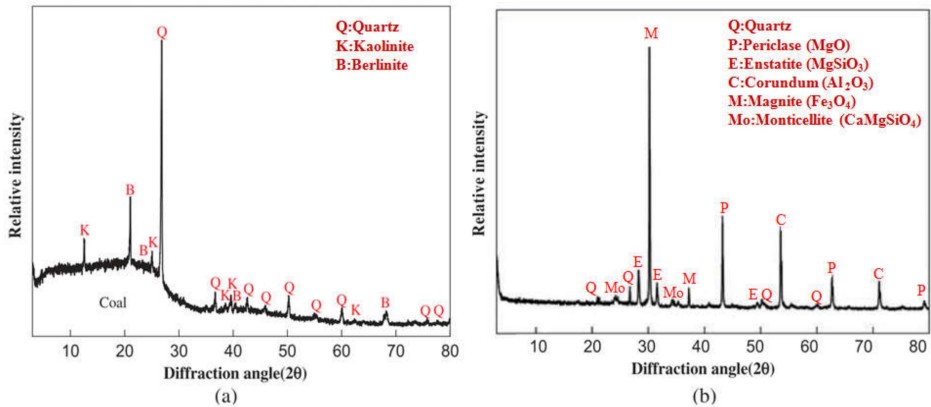

**Figure 7.** Wavelength dispersive X-ray diffraction spectra for (**a**) pre-explosion and (**b**) post-explosion for Laohutai coal dusts [257].

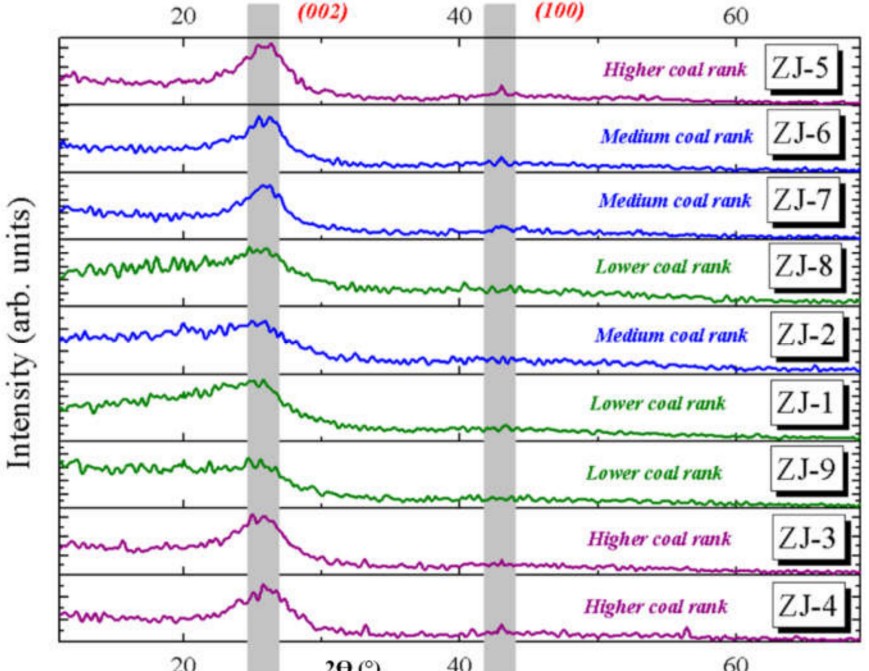

**Figure 8.** Wavelength dispersive X-ray diffraction profiles for demineralized coal samples [260].

Crystalline carbon (anthracite), a graphite-like structure, has a Π-band (002) peak at ~26 and (100) peak at ~42°. The asymmetric Π-band (002) peak shown in coal samples suggests the existence of another γ-band peak at ~20°, referenced to the saturated structures such as aliphatic side-chains [260,272,273].

### 4.2.5. Fourier-Transform Infrared (FTIR) Spectroscopy

FTIR detects functional groups (e.g., alcohols, amines, carboxylic acids, and ketones) and has been widely applied to examine the chemical structure of coal. This is a non-destructive method that identifies molecular vibrations (both stretching and bending) depending on the infrared radiation absorption [274,275]. The sample is exposed to continuous oscillation of different infrared (IR) wavelengths, where IR is absorbed when the incident radiation is equivalent to the energy of a particular molecular vibration. Wave numbers ranging from 1200 cm$^{-1}$ to 4000 cm$^{-1}$ correspond to the stretching vibration energy, while the matching wave numbers for bending vibrations are 500 cm$^{-1}$ to 1200 cm$^{-1}$. An example of band assignment for bituminous coal is shown in Table 5. The presence

of functional groups can be detected in the absorption peaks. Recently, NIOSH focused on evaluating FTIR to quantify EC and OC in filter samples derived from diesel exhaust emissions and from mine air samples. Preliminary data showed that the FTIR direct-on-filter (using PVC filter) method may be useful for DPM quantification, along with multi-variate analysis of the spectrometry data, to estimate the EC and OC in airborne diesel emissions [276].

**Table 5.** Band assignments of bituminous coal derived from FTIR Spectra [260].

| Band Wave Number (cm$^{-1}$) | Functional Groups | Peak Intensity | | |
|:---:|:---:|:---:|:---:|:---:|
| | | **L** | **H** | **A** |
| 3419–3359 | –OH stretching vibration | W | S | S |
| 3080–3035 | Aromatic CH stretching vibration | S | M | W |
| 2975–2848 | Aliphatic CH stretching vibration | M | S | W |
| 1745–1695 | C=O | S | M | W |
| 1615–1585 | C=C | S | M | W |
| 1500–1450 | C–C stretching | W | M | S |
| 1300–1000 | C–O–C stretching | S | M | W |
| 900–700 | C–H out–plane bending | S | M | W |

L: low-rank bituminous coal; H: high-rank bituminous coal; A: Anthracite; S: strong; M: medium; W: weak.

Overlapping bands often occur in IR spectroscopy, which can be addressed by enhancing the resolution by derivative spectroscopy and Fourier self-deconvolution methods. This method has been applied to determine organic functional groups, such as aliphatic hydrogen, aromatic hydrogen, and oxygen-containing groups in coal samples [272,277].

FTIR has gained wider application since both organic functional groups [278–280] and mineral composition can be analyzed (e.g., clay, sulfate, slag, pottery, and oil shale) [281–283]. The advantages of FTIR over XRD in identifying minerals include (1) identifying both crystalline and amorphous phases, (2) distinguishing the origin of water molecules (e.g., structural water, coordinated water, and zeolitic water), and (3) performing rapid and less costly analysis [255].

FTIR spectra derived by subtracting the absorption spectrum of demineralized coal from that of raw samples can identify mineral matter [284]. Table 6 shows the absorption bands of minerals in coal samples [255]. Mukherjee and Srivastava [285] found the band shift of mineral transformation in coals from kaolinite (1025 cm$^{-1}$) to quartz (1081 cm$^{-1}$) occurs at 850 °C due to heating. Similarly, Bai et al. [286] studied the coal ash FTIR spectra at 1300 and 1400 °C under reducing conditions. They reported that the strongest absorption bands are associated with asymmetric Si–O–Si or Si–O–Al stretching vibrations of aluminosilicates in the range of 1100–950 cm$^{-1}$. Mozgawa et al. [287] identified amorphous aluminosilicates in fly ash by FTIR at 915 cm$^{-1}$, which are not detected by XRD. Han et al. [288] found kaolinite explaining the relatively high $SiO_2$ and $Al_2O_3$ content in lignite coals. The least squares curve-fitting method by Painter et al. [289] shows comparable mineralogical compositions (e.g., kaolinite, quartz, calcite, pyrite, and illite) in coal dust. The FTIR method has also been used to measure the quartz content in coal ashes [290].

Both MSHA [291] and NIOSH [292] have published standard methods to measure quartz and kaolinite contents in coal dust. The MSHA method identifies the absorbance spectra with a baseline of 815 to 770 cm$^{-1}$ for quartz and 930 to 900 cm$^{-1}$ for kaolinite, whereas the NIOSH 7603 method specifies the absorbance peak at 800 cm$^{-1}$ with a baseline of 820 to 670 cm$^{-1}$ for quartz and at 915 cm$^{-1}$ with a baseline of 960 to 860 cm$^{-1}$ for kaolinite. Calibration curves were developed based on known masses (from an aliquot volume of suspension in isopropyl alcohol) of respirable $\alpha$-quartz and FTIR absorbance at 800 cm$^{-1}$.

**Table 6.** FT-IR absorption bands of mineral matter in coal samples [255].

| Mineral | FTIR Absorption Bands (cm$^{-1}$) |
|---|---|
| Anhydrite | 1154, 1120, 679, 613, 595 |
| Quartz | 1164 [a], 1082 [a], 797, 778, 696 [a], 513 |
| Calcite | 1797, 1447 [a], 875, 713 |
| Aragonite | 1476, 857 |
| Microcline | 646, 534 [a] |
| Albite | 425 [a] |
| Amorphous silica | 1099 [a], 1013 [a] |
| Metakaolinite | 1030 [a], 562 [a] |
| Portlandite | 3641 |
| Nitrate | 1385 |
| Unknown aluminasilicate | 479 [a], 445a |

[a] overlapping bands.

The standard method to quantify silica in coal mines involves collecting particles on a PVC filter for laboratory analysis by the MSHA P-7 [293] or NIOSH 7603 [292] methods. These methods are labor-intensive (~1–2 weeks) and do not provide timely results relevant to exposure control and worker protection. NIOSH developed a direct-on-filter silica quantification method using portable FTIRs that can provide end-of-shift measurements on site [294,295]. However, interferences from other minerals can lead to inaccurate silica quantification [296,297]. Spectrum deconvolution and correction algorithms are still under development.

### 4.2.6. Raman Spectroscopy

Raman or micro-Raman spectroscopy determine minor or trace impurities in the chemical composition and morphological structure of a mineral sample. Raman and XRD spectra for minerals can be found in the "RRUFF" database [298]. The various shapes of Raman bands are used to attribute sharp bands for crystalline minerals and broad bands for amorphous phases or fluorescence. Although Raman spectroscopy has been widely applied for characterizing mine waste [299], cement [300], and pottery [301], limited research has targeted coal dust characterization [302,303].

Both XRD and FTIR techniques have been used to measure respirable crystalline silica in dust collected on a filter [304] with limits of detection (LOD) between 3 and 10 μg [304]. Stacey et al. [305] found quantification limits of 0.066–0.161 μg for quartz and 0.106–0.218 μg for cristobalite at 464 cm$^{-1}$ and 410 cm$^{-1}$, respectively, by Raman microscopy using 5 mm diameter silver filters. Zheng et al. [306] employed a field-portable Raman spectrometer and achieved comparable LODs in the range of 0.008–0.055 μg for quartz with a sample spot diameter of 400−1000 μm. These LODs can be lowered with longer collection times at various locations. Raman spectroscopy can be effectively used to differentiate polymorphs and microcrystalline silica [307]. It is also implemented in identifying the minerals composing the rocks found on moons and other planets and in the pharmaceutical industry [308].

The advantages of Raman spectroscopy include (1) in situ, non-destructive analysis without sample preparation required; (2) low detection limit; (3) micrometer-scale characterization; (4) versatility in detecting amorphous compounds; and (5) potential for on-line coal dust characterization.

Applying Raman spectra for coal was first introduced by Tuinstra and Koenig [309] and Friedel and Carlson [310]. Two major bands were found in the regions 1575–1620 and 1355–1380 cm$^{-1}$, called the G (graphitic) and D (disordered) bands, respectively. The 1580 cm$^{-1}$ band was assigned to the $E_{2g}$ graphite mode with $D_{6h}^4$ crystal symmetry,

and the 1370 cm$^{-1}$ band was assigned to the A$_{1g}$ mode [309]. Friedel and Carlson [310] investigated finely ground graphite (broken C–C bonds), coal, and carbon black, and related the 1370 cm$^{-1}$ band to graphitic structures. Figure 9 shows the Raman spectra for a laboratory-generated sample consisting of 80% coal and 20% quartz that indicates the D, G, and quartz bands. Potgieter-Vermaak et al. [308] characterized oxides, sulfides, and silicates and identified calcite, pyrite, and dolomite as inorganic matter in coal.

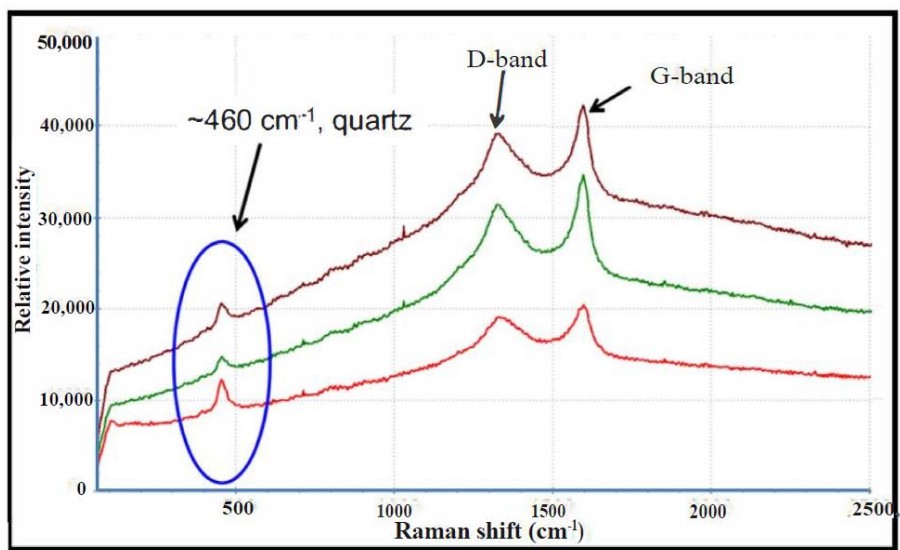

**Figure 9.** Raman spectra for laboratory-generated dust samples consisting of 80% coal and 20% quartz.

Raman spectroscopy has been used to characterize cement, marine aerosol, and re-suspended dust (with 532 nm laser) [300,311]. Stacey et al. [312] shows that the size of particles does not alter the Raman responses for the mass of 0.25–10 μg samples.

### 4.2.7. $^{13}$C and $^{1}$H Nuclear Magnetic Resonance (NMR) Spectroscopy

$^{13}$C NMR spectroscopy is a non-destructive analysis that can identify chemical structures in coal [313]. This technique has not yet been applied to RCMD. Coal is a matrix of aromatic clusters with aliphatic and carbonyl side chains and solvent-extractable components [314]. $^{13}$C NMR can determine lattice structures of coal with the following parameters: number of carbons and attachments per cluster, total number of bridges and loops, ratio of bridge to total attachments, average aromatic cluster molecular weight, and average side chain molecular weight [314]. Retcofsky et al. [315] showed that the $^{1}$H NMR method can determine the aromaticities of coal-derivatives.

### 4.2.8. Example of Comprehensive RCMD Chemical Characterization

Due to the complexity and heterogeneity of RCMD, multiple analyses are needed to provide a comprehensive characterization. Accordingly, multiple filter media are needed for a comprehensive chemical speciation [238]. Chow and Watson [240] proposed that three parallel channels, including a Teflon membrane filter, a quartz fiber filter, and polycarbonate membrane filter, would provide options for dust sampling. Using the same strategy, Table 7 outlines an example of analyses that can be applied to achieve quantitative and qualitative chemical characterization of RCMD.

**Table 7.** Comprehensive chemical and morphological analyses for Teflon membrane, quartz fiber, and polycarbonate membrane filter substrates.

| Channel 1 Teflon Membrane | Channel 2 Quartz Fiber | Channel 3 Polycarbonate |
|---|---|---|
| **XRF** (elemental analysis) Function: identify wide variety of elements (51 elements Na to U) Limitation: low concentrations of several rare-earth elements (lanthanide series) or light elements (Li, Be, and B) cannot be identified. | **TOR/TOT** (carbon analysis) Function: identify OC, EC, brown carbon (BrC), and carbonates Limitation: destructive process and uncertainty in char correction that separates EC from OC. | **SEM-EDX**: (morphological and elemental analysis) Function: size and shape analysis; Identify elements with atomic number larger than ~12 Limitation: captures only a small fraction of particles; labor intensive |
| **ICP-MS** (elemental analysis) Function: complement XRF with additional rare-earth elements and with lower minimum detection limits Limitation: destructive method; preparation and sample extraction may contaminate the sample or lead to incomplete analysis. | **TD-GC-MS** (organic molecules analysis) Function: quantify ~110 non-polar organic compounds, including alkanes, alkene, hopanes, steranes, and PAHs Limitation: destructive process and only a fraction of organic compounds are analyzed. | **FTIR**: (chemical composition analysis) Function: identify organic functional groups and mineral composition (including quartz) for both crystalline and amorphous states Limitation: overlapping bands often occur in infrared spectroscopy; minimum detection limit is high to detect quartz (between 3 and 10 μg) |
| **XRD** (mineralogy) Function: composition and structure of crystal components (e.g., gypsum, and metal oxides) Limitation: high minimum detection limit (between 3 and 10 μg); cannot characterize disordered materials quartz | | **Raman Spectroscopy**: (chemical composition analysis) Function: complement FTIR results with lower minimum detection limit (crystalline silica and spectral fingerprints) Limitation: not suitable for particles with high absorption |

XRF: X-ray fluorescence. ICP-MS: inductively coupled plasma-mass spectrometry. XRD: X-ray diffraction. TOR-TOT: thermal/optical analysis by reflectance and transmittance. TD-GC-MS: thermal desorption-gas chromatography-mass spectrometry. SEM-EDX: scanning electron microscopy with energy dispersive X-ray detection. FTIR: Fourier-transform infrared spectroscopy.

To determine total RCMD mass, the Teflon membrane filter is weighed before and after sampling. Teflon filters are then submitted for elemental analysis of 51 elements (Sodium [Na] to Uranium [U]) using XRF [240]. Since dust particles do not penetrate deeply into the Teflon membrane, Teflon membrane filters are preferred over fibrous filters. Next, mineral contents (e.g., gypsum, and metal oxides) are determined using XRD. As these are non-destructive analyses, the same Teflon filters are then submitted to hot block acid extraction for the analysis of additional elements by ICP-MS. Half of the quartz fiber filter can be acidified by hydrochloric acid to measure carbonate, and then analyzed for organic and elemental carbon (OC/EC) by the TOR/TOT method following the NIOSH Method 5040 [220]. Another portion of the quartz fiber filter is submitted for organic speciation, including alkanes, cycloalkenes, polycyclic aromatic hydrocarbons (PAHs), hopanes, and steranes. The polycarbonate filter would be submitted for size, morphology, and elemental analysis by SEM with an EDX detector. A subset of these filters can be analyzed for crystalline silica using the FTIR and Raman spectrometers. The FTIR can also analyze organic functional groups.

## 5. Summary and Conclusion

Although adverse effects of RCMD on workers' health have been recognized for decades and several regulations and research efforts have been focused on this issue, there is an increasing prevalence and severity of coal mine dust-related lung diseases in some regions. This review assesses measurement technologies that characterize coal mine dust mass concentrations, size distributions, and chemical/mineral constituents for mining areas. Comparisons of different techniques are summarized with examples where these methods have been applied (with a focus on U.S. coal mines). Some of the

advanced instrument presented in this paper are not intrinsically safe (e.g., ELPI, APS, and AAC) and caution should be exerted when using them in explosive environments. Outlines for performing comprehensive characterization of RCMD size distribution and chemical composition are recommended. This review indicates that many coal mine dust size distributions are decades old and may not represent modern mining technologies (e.g., increased equipment size and power and mining thinner coal seams). It is apparent that RCMD and silica exposures need to be supplemented with more detailed chemical knowledge of potentially toxic species. Future studies are essential to provide insights into the causes for recent increases in coal miner lung diseases.

**Supplementary Materials:** The following are available online at https://www.mdpi.com/article/10.3390/min11040426/s1, Table S1: Summary of Respirable Coal Mine Dust (RCMD) characterization studies, Table S2: Summary of Respirable Coal Mine Dust (RCMD) chemical characterization studies.

**Author Contributions:** Conceptualization, B.A., X.W., J.G.W. and J.C.C.; investigation, M.E., B.A. and X.W.; writing—original draft preparation, M.E., B.P. and K.B.R.; writing—review and editing, X.W., J.G.W., J.C.C. and B.A.; visualization, V.N.; supervision, B.A.; project administration, B.A. and X.W.; funding acquisition, B.A. All authors have read and agreed to the published version of the manuscript.

**Funding:** This research was funded by National Institute for Occupational Safety and Health (NIOSH), grant number: 0000HCCS-2019-35133.

**Acknowledgments:** We acknowledge that many relevant studies could not be cited due to the space restrictions.

**Conflicts of Interest:** The authors declare no conflict of interest. The funders had no role in the design of the study; in the collection, analyses, or interpretation of data; in the writing of the manuscript, or in the decision to publish the results.

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
