# Peer review of "Review of Respirable Coal Mine Dust Characterization for Mass Concentration, Size Distribution and Chemical Composition"

_minerals, doi:10.3390/min11040426_

Round 1

Reviewer 1 Report

see attached. 

Author Response

Dear Reviewer,

Thanks for the comments. We have addressed all your comments in the revised manuscript. Please see the attached file for more detail. 

Thanks,

BA

Reviewer 2 Report

This manuscript reviews particulate matter measurement methods for mass concentration, size distributions, and chemical composition. What makes this review of interest (there are many such reviews in the peer reviewed literature) is that is specifically addresses applications of such methods for characterizing exposures of coal miners to respirable coal mine dust.

The manuscript is well written; I can only offer a few very minor edits, as follows:

lines 51-52: "... the prevalence and severity... HAVE increased.." (change "has" to "have")

lines 91-92: "...size distributions were collected over a decade ago and NO LONGER REPRESENT..." (remove the word "are")

lines 104-105: "RCMD contains transition metals such as iron..." I would suggest this needs references.

Table 1, first row, third column, "Data ARE not immediately available" (change "is" to "are")

Figure 2b caption, "Thermo TEOMM Personal Dust Monitor..." - Should this be "ThermoScientific Personal Dust Monitor..."?

Line 151: "MSHA (2014) requires the use..." - this is not consistent with citation formatting throughout remainder of paper.

Line 153: "...(Part 90 miners)." This is also not consistent formatting.

Line 580: "Several researches..." this is probably personal preference, but should this be "Several research projects..."?

Section 4: At the risk of recommending another (big) table to an already long manuscript, this section on chemical composition may benefit from a table (or two) comparing methods, similar to tables 1 and 2, in sections 2 and 3.

Author Response

(The authors gave the same response as above.)

Reviewer 3 Report

Dear Authors,

Thank you for putting this manuscript together. It is important to have a review paper on  coal mine dust characterization. You can find my comments below:

-As far as I know, researchers generally use PVC type filter with FTIR. Please include that in your manuscript.

-Coal mine dust characterization is not limited with the samples on filters. Doctors are also characterizing coal mine dust in lung tissue biopsies using SEM ,optical microscope, ICP etc. Please include those studies in your manuscript as well.

Author Response

Thanks for the comments of yours and the reviewers. We have addressed all your comments and the reviewers’ in the revised manuscript.

Reviewer 4 Report

Minerals 1138806 review

Several instrumentations and monitors described and mentioned in the article might not be intrinsically safe to be used in most parts of a coal mine. For example – ELPI, MOUDI, APS, SMPS, OPC, FMPS. The authors should mentioned this limitation clearly in the abstract and conclusions

Line 73 – the ACGIH adopts the inhalable, thoracic, and respirable convention in terms of particle size distribution as defined by (ISO/CEN/ACGIH). It is a misleading approximation that all particles smaller than 4 um are respirable particles. The convention is characterized by the probability of a particle of a certain size to be respirable. A particle of 10 um, for example has 10% probability to be respirable. In comparison a particle of 3 um might not have a probability larger than 50%. Please refine.

Line 83 – the word “crude” approximation is offensive to decades of work on respirable dust sampling. Every size selector has known bias towards the respirable convention. Please refine.

Table 1 – There is no known laboratory analysis for the filter media currently used in the CPDM. It is incorrect that this is an option.

Line 144 – the verb “distinguish” should be replaced with “appreciate” or “record”.

Line 180 – the “transport losses and particle blow-off” studies need to be referenced or the statement removed.

Line 189 – why the cost of the unit can prevent a company to use the CPDM for other activities that are not regulatory? If a company has a CPDM, why not using it?

Line 224- I am sure there are more than two references for measurements of particle size distributions. What is special about [51] and [52]. Please revise.

Line 344 – Is the sampling pump of a MOUDI intrinsically safe? This should be verified.

Line 394 – it should be noted that Bugarski did not used an ELPI in a coal mine, but a trona mine. Please revise.

Table 3 – Consistency issue – why SiO2 is the only mineral presented with a formula?

Line 602 – the word “concentration” is incorrect. The study [174] focused on silica content as percent of quart in respirable dust. Please revise.

Line 893 – Consistency with how a reference (Stacey) is reported.

Author Response

(The authors gave the same response as above.)
